# Effects of Sodium Alginate, Pectin and Chitosan Addition on the Physicochemical Properties, Acrylamide Formation and Hydroxymethylfurfural Generation of Air Fried Biscuits

**DOI:** 10.3390/polym14193961

**Published:** 2022-09-22

**Authors:** Mingchih Fang, Yen-Shu Ting, Wen-Chieh Sung

**Affiliations:** 1Department of Food Science, National Taiwan Ocean University, Keelung 202301, Taiwan; 2Center of Excellence for the Oceans, National Taiwan Ocean University, Keelung 20231, Taiwan

**Keywords:** acrylamide, biscuit, reducing sugar, sodium alginate, pectin, chitosan, air frying

## Abstract

This study evaluated the effects of sodium alginate, pectin and chitosan addition (0.5–1.5%) on the physicochemical properties including pH, water activity, moisture content, color values, hardness, diameter, thickness, spread ratio, antioxidant activities and sensory scores of biscuits in air frying processing. In addition, the formation of acrylamide and hydroxymethylfurfural (HMF) were discussed. Physicochemical properties of biscuits including water content, water activity, hardness, appearance, shape, color, flavor, texture, overall acceptability, and DPPH radical scavenging activity of biscuits were not influenced significantly by the addition (0.5–1.0%) of three food hydrocolloids. The data showed that the biscuits with hydrocolloids addition had lower acrylamide contents than that of the control biscuit without hydrocolloids addition, and the reducing power of biscuits increased after adding the hydrocolloids. The highest mitigation of acrylamide formation was obtained by the chitosan addition formulation. The formation of acrylamide showed a negative correlation with the content of sodium alginate and chitosan addition, and they were effective ingredients in terms of mitigating the formation of acrylamide in biscuit formulation.

## 1. Introduction

The Caramelization and Maillard reaction produces a desirable color and flavor during the baking process of the biscuit. However, there are also potential mutagenic- and genotoxic-causing compounds such as acrylamide and hydroxymethylfurfural (HMF) generated during baking [1,2,3]. Acrylamide formation is associated to fats, sugars and amino acids in food which are heated above 120 °C to trigger Schiff base generation and Strecker degradation [4,5]. Then, the derived carbonyl-containing compounds such as acrolein (CH_2_=CH-CHO, the fatty acid oxidation product) react with ammonia-containing compounds such as mainly *N*-glycosides and asparagine to form acrylamide via oxidation [6,7,8].

The baking industry has been looking for proper ingredients and methods to mitigate acrylamide and HMF formation in bakery products [9]. Jung et al. [10] proposed that the conversion of free non-protonated amine into non-nucleophilic protonated amine by lowering the pH in a food system was an efficient method to decrease acrylamide formation. It has been demonstrated in cookies, potato chips, corn chips and French fries by adding organic acids to lower the pH levels which has showed mitigating effects on acrylamide [10,11,12]. Other strategies were reported in the last two decades, such as adding calcium salts [13,14], using non-reducing sugar such as sucrose instead of reducing sugars including glucose and fructose [15,16,17], applying asparaginase to degrade free asparagine in food systems [18,19,20,21], adding glycine to dilute the asparagine level [22], soaking and blanching potato strips to remove the acrylamide precursor and decrease frying time [23], preventing the use of ammonium containing baking powder [24].

Polysaccharides were reported as good acrylamide inhibitors in food systems such as chitosan, pectin and sodium alginate. Chitosan is a deacetylated derivative ((1-4)-2-deoxy-β-d-glucan) of chitin from the exoskeletons of crustacean shellfish, e.g., shrimp and crab. The deacetylated chitosan contains amino groups which are able to react with the carbonyl groups of reducing sugar to form Maillard reaction products [13,25]. Pectin was reported able to compete between amino acids and reducing sugars to mitigate acrylamide formation [26]. The combinations of sodium alginate and pectin solution (0.2–0.3%) were shown to be more effective on the inhibition of acrylamide formation in fried potato chips [27]. Among acrylamide mitigating methods mentioned above, the addition of polysaccharides seems to be a good way in the biscuit manufacturing. Biscuit is made of ingredients including flour, water, sugar, shortening, salt, baking powder, non-fat dry milk, and emulsifier [28,29,30]. The application of chitosan, pectin and sodium alginate into the ingredients list of biscuit would not alter the formula too much but may decrease the formation of acrylamide.

The air fryer is widely used in home cooking operations which is an innovative food cooking approach to reduce the oil content of homemade foods and snacks while maintaining acceptable quality [31,32]. The air fryer uses superhot air circulation rather than hot oil, offering a shorter cooking time than conventional oven baking [33]. This new frying method was applied to biscuit baking processing. There is no report on the technological considerations based on air frying biscuits. The physical and chemical reactions related to appearance, dimension, texture, taste, acrylamide and HMF formation during air frying needed to be investigated in order to help the food industry develop better strategies for acrylamide mitigating. This study developed biscuit recipes with sodium alginate, pectin or chitosan additions (0.5–1.5%) for air frying and investigated the role of hydrocolloids in the mitigation of acrylamide. The physicochemical characteristics of the biscuits were also evaluated to develop better product quality and lower acrylamide content of air frying products.

## 2. Materials and Methods

### 2.1. Chemicals and Ingredients

Cake flour used in this study was supplied from the Cha Hwa Corporation (Taichung, Taiwan). The cake flour contains moisture 13.0%, protein 8.0%, fat 1.0%, ash 0.4%, and total carbohydrate 77.6%. Nonfat dry milk, shortening, sodium chloride, and sucrose were purchased from Happiness Food Ingredient Store (New Taipei City, Taiwan). Food grade chitosan was obtained from Charming & Beauty Co., Ltd., Taipei City, Taiwan. Commercial citrus pectin powder containing glucose (1/4) was obtained from Tehmag Foods Corporation, New Taipei City, Taiwan. Sodium alginate was purchased from Shun-Chin Raw Material Co. Ltd. Kaohsiung City, Taiwan.

Acetic acid, 1,1-dihpenyl-2-picrylhydrazyl (DPPH) and ferrozine were supplied by Sigma Aldrich (St. Louis, MI, USA). Potassium sodium tartrate, 3,5-dinitrosalicylic acid, sodium hydroxide, and sulfuric acid were purchased from Panreac (Barcelona, Spain). Acrylamide 99.9% was supplied from J.T. Baker (Phillipsburg, NJ, USA). Oasis HLB (6 mL, 0.2 g) and Oasis MCX (3 mL, 0.06 g) solid phase extraction (SPE) cartridges were from Waters (Milford, MA, USA). Sodium hydroxide, trichloroacetic acid, potassium ferricyanide, ethylenediaminetetraacetic acid disodium salt dehydrate (EDTA), ferrous chloride and ascorbic acid were from Merck (Whitehouse Station, NJ, USA). Hydroxymethylfurfural was supplied by Acros Organics (Fair Lawn, NJ, USA). Ferric chloride was from Hanawa Chemical Pure Corporation (Tokyo, Japan). All chemicals used in this work were of analytical grade.

### 2.2. Biscuit Preparation and the Measuring of Physicochemical Properties of Biscuits

The biscuit formulation referred to by the American Association of Cereal Chemists (AACC) method 10–54 (80.0 g cake flour, 17.6 g deionized water, 25.6 g sucrose, 0.8 g non-fat dry milk, and 32.0 g shortening) was slightly modified [34]. Fructose syrup was replaced with sucrose. Chitosan, pectin or sodium alginate (0% (control), 0.5% (0.4 g), 1.0% (0.8 g) or 1.5% (1.2 g)) of the cake flour weight (80 g) was used in the biscuit’s formulation. The biscuit dough was rolled out to a thickness of 5 mm and cut with a circular biscuit cutter (diameter: 60 mm). Four biscuit doughs were air fried in a commercial hot air fryer (HD 9642, Philips, Kaohsiung City, Taiwan) at 170 °C for 15 min. Biscuits were cooled down at room temperature for 2 hrs, and the biscuits were stored in polyethylene (PE) bags at −20 °C freezer until use.

The diameter and thickness of biscuits were measured based on the average of ten measurements by a Venier Caliper. The biscuit spread ratio was calculated from dividing the diameter by the thickness of the biscuit [35]. The effect of hydrocolloids on the pH of the biscuits was determined by the method of Navarro and Morales [36], using a pH meter (pH meter, Thermo Eutech pH/Ion 510, Singapore). The water activity of the biscuit was measured using a dew point water activity meter AquaLab CX-2 (Smartec Scientific Corp., New Taipei City, Taiwan). The moisture content was obtained by the mass loss of biscuit (around 2.0 g) in an oven at 105 °C to a constant weight [37].

### 2.3. Acrylamide and HMF Levels in Biscuits

Biscuits were ground with a pulverizer (D3V-10, Yu Chi Machinery Co., Ltd., Chang Hua, Taiwan). Sample 1 g each was weighed and put into a 15-mL centrifuge tube, 9 mL of deionized water and 1 mL hexane was added. The tube was placed in a 25 °C reciprocal shaker bath for 60 min. The mixture was centrifuged at 5 °C for 20 min at 1500× *g*. The supernatant (3 mL) was filtered through a nylon filter (0.45 μm) and passed through a HLB/MCX cartridge which was pre-conditioned with 5 mL and 3 mL methanol, followed by 5 mL and 3 mL of deionized water, respectively. The filtrate (3.0 mL) was passed through an Oasis HLB/MCX cartridge, and the filtrate was discarded. The cartridge was then eluted with 3.5 mL deionized water, the first 0.5 mL filtrate was discarded, and the remaining eluent was collected and transferred into a brown glass tube. The eluent was concentrated under a vacuum prior to HPLC analysis [38,39].

The HPLC system (LC 2040C 3D Plus) consisted of a column oven, pump, detector and autosampler (Shimadzu Corporation, Kyoto, Japan). Chromatographic separation was achieved by a Capcell Pak C_18_ AQ S5 column (5 μm, 4.6 mm × 250 mm) (Shiseido, Tokyo, Japan) using isocratic elution of deionized water at a flow rate of 0.7 mL/min at 25 °C. The injection volume was 20 μL. The acrylamide was monitored by using a UV detector at 210 nm, and the calibration curve was prepared in the range of 0–3125 ppb.

HMF analysis followed using the method of Oral et al. [40] with some modifications. The sample extract was purified by a HLB/MCX cartridge similar to the sample preparation of acrylamide. Chromatographic separation was completed via a Capcell Pak C_18_ AQ S5 column (5 μm, 4.6 mm × 250 mm) (Shiseido, Tokyo, Japan) using isocratic elution consisting of deionized water, formic acid and acetonitrile (94:1:5) at a flow rate of 1.0 mL/min at 25 °C. The autosampler was maintained at 10 °C, and the inject volume was 20 μL. HMF standard solutions for calibration curve were prepared in the range of 1–100 ppb and monitored at 284 nm.

### 2.4. DPPH Radical Scavenging Assay, and the Determination of Reducing Power

The ground sample (5 g) each was extracted with 45 mL of ethanol in a centrifuge tube. The tube was placed into a reciprocal shaker at 25 °C and shook at 100 rpm for 24 h, according to the method described by Kim and Rajapakse [41] with slight modification. The extract was centrifuged (1000× *g*) at 4 °C for 30 min and filtered with filter paper.

The scavenging effects of biscuit samples for DPPH were evaluated spectrophotometrically according to the methods of Shimada et al. [42]. A 0.5 mL aliquot of biscuit extracted solution was mixed with 0.5 mL of 0.1 mM DPPH ethanolic solution (95%). The solution was kept at room temperature for 30 min in the dark. The absorbance was determined at 517 nm and the percentage of the radical scavenging effect was calculated using the following equation:Scavenging effect (%) = (1 − (A_sample_/A_blank_)) × 100%
where A_sample_ is the absorbance of the biscuit sample (DPPH plus extracted solutions) and A_blank_ is the absorbance of water plus ethanolic DPPH solution.

Reducing power was determined spectrophotometrically using the ferricyanide method by Kanatt et al. [43]. Biscuit extract (1 mL) was mixed with 1 mL of sodium phosphate buffer (0.2 mM, pH 6.6) followed by 1 mL of 1% potassium ferricyanide. The solution was placed in a water bath at 50 °C for 20 min. After incubation, the tube was cooled in ice and 1 mL of 10% trichloroacetic acid was added and mixed. The mixture (1 mL) was added of 1 mL deionized water and 0.2 mL of 0.1% ferric chloride solution and kept at room temperature for 10 min. Absorbance of the resulting solution was recorded at 700 nm.

### 2.5. Hardness and Color Measurement of Biscuits

Hardness of biscuit was analyzed by the method of Passos et al. [44]. A TA-XT2 Texture Analyzer with a round probe (P/25A, 4 mm in diameter) was accomplished to compress biscuit samples at 1.00 mm/s to for 15 mm (trigger load: 20.0 g; target distance: 15 mm; supports apart: 50 mm). The maximum force during compression was determined. The test was performed in triplicate and the average maximum force was recorded as hardness in Newton (N).

The color of the biscuit was evaluated with a spectrocolorimeter (TC-1800 MK II, Tokyo, Japan) by the L* (lightness), a* (redness/greenness) and b* (yellowness/blueness) color scale. A standard white tile was used to calibrate the spectrocolorimeter [44]. Biscuit samples were loaded onto a quartz sample cup with three measurements for each sample, and triplicate determinations were evaluated per treatment. The brown index (BI) and color difference were calculated using the following equations: BI = 100 × [x − 0.31]/0.17
where x = (a* _sample_ + 1.75 L* _sample_)/(5.645L* _sample_ + a* _sample_ −3.012b* _sample_).
The color difference ΔE = [(ΔL)^2^ + (Δa)^2^ + (Δb)^2^]^1/2^
where ΔL = L_sample_ − L_control_; Δa = a_sample_ − a_control_; Δb = b_sample_ − b_control_.

### 2.6. Sensory Evaluation of Air Frying Biscuits 

A panel consisting of thirty-five male and forty-two female students from the Department of Food Science within the age range of 20 to 29 were recruited. Biscuit samples were coded with three random digits and panelists were instructed to score the appearance, texture, flavor, odor, and overall acceptability using a nine-point hedonic scale ranging from “1 = extremely dislike” to “9 = extremely like” according to the method of Sudha et al. [45].

### 2.7. Statistical Analysis

Differences of the means were determined using Duncan’s Multiple Range Test. Data were evaluated by the analysis of variance and Duncan’s new multiple range test using SPSS 2000 statistics program for Windows, Version 12 (SPSS Inc., Chicago, IL, USA) at 95% significance level (*p* < 0.05). Linear correlations of different physicochemical properties and the test data were evaluated by Pearson’s correlation at a significance level of 0.05.

## 3. Results

### 3.1. Physicochemical Properties of Biscuits

Physical characteristics of biscuit diameter, thickness and spread ratio were not affected by the addition of food hydrocolloids (Table 1). The thickness of biscuits made with food hydrocolloid addition was in the range of 8.15 mm to 8.67 mm, which were not statistically significant to the thickness (8.54 mm) of the control biscuit (without the addition of hydrocolloids). The diameter of biscuits made with hydrocolloid addition was in the range of and 56.54 mm to 58.80 mm, which was also not statistically significant to the diameter (57.39 mm) of the control biscuit. Moreover, the spread ratio was between 6.62 and 7.18 for hydrocolloid-added biscuits, which was also not different (*p* > 0.05) to the spread ratio (6.74) of the control biscuit. A higher cookie spread ratio and diameter were thought as the preferred cookie characteristics [46]. The pH values of the biscuits were in the range of 5.53 to 6.01. The lowest pH was observed in biscuits with 1.0% and 1.5% commercial pectin powder addition (pH 5.54 and 5.53, respectively) comparing to the control biscuit (pH5.9) (Figure 1). The moisture content of the biscuits was in the range of 1.2−2.0%. In addition, a_w_ of the biscuits was in the range of 0.19–0.23 after 15 min air frying.

### 3.2. Acrylamide and Hydroxymethylfurfural Levels in Biscuits

The acrylamide concentrations in the control biscuit and hydrocolloid-added biscuits were 1340 ppb and between 578 and 965 ppb (Figure 2A). For all the biscuits, HMF was not detected except for the 1.0% and 1.5% commercial pectin-powder-added samples (1.31 ppb and 1.45 ppb, Figure 2B).

### 3.3. DPPH Radical Scavenging and Reducing Power of Biscuits

There was no significant difference among control and hydrocolloid-added biscuits on DPPH radical scavenging activities (Data not shown). For reducing power, all of the hydrocolloid-added biscuits except the 0.5% sodium alginate addition, showed an increased reducing power (*p* < 0.05). The pectin-added biscuits (with 0.5% and 1.5% pectin) were observed as having the best reducing power among all samples (Figure 3).

### 3.4. Hardness, Color and Sensory Evaluation of Biscuits

The addition of food hydrocolloids did not lead to a significant change in the hardness of the biscuits (Figure 4). Adding 0.5% sodium alginate gained the hardness of the biscuit around 36.8 N, which showed no difference to the hardness of the control biscuit (43.4 N).

The color and appearance of the biscuits play an important factor in consumers’ perception and acceptability of the biscuit. There was no significant difference between groups. The values of L*, a* and b* were found to be 65.2–70, −1–2, and 49–57, respectively (Table 2 and Figure 5).

The results for appearance, odor, flavor, texture and overall acceptability for the selected samples, which offered the best acrylamide mitigating effects, were shown in Table 3. The appearance and odor of the control biscuit was rated higher (7.05 and 7.04, respectively) than biscuits with 0.5% pectin and 0.5% chitosan additions (Table 3; *p* < 0.05). There was no significant difference among all groups in the flavor, texture, and overall acceptability. Although sensory scores of the control were slightly higher than those of hydrocolloid addition biscuits, they were still in an acceptable range (>5).

## 4. Discussion

### 4.1. Physicochemical Properties of Biscuits

The ingredients in cookies contain more sugar and flavoring agents compared to biscuits which are mainly made using butter and flour. Therefore, cookie dough is slightly heavier and less fluffier than biscuits. Biscuits require hard dough, whereas cookies require soft dough. In this study, three hydrocolloids, chitosan, pectin and sodium alginate, were added (0.5–1.5%) in the formulation of biscuits. The changes in diameter, thickness and spread ratio were not significantly different from the control biscuit (Table 1). These results indicate that the addition of hydrocolloids did not influence the biscuit dough rheology.

The pH of the biscuits decreased with the addition of 1.0% and 1.5% pectin (Figure 1). This may be due to the commercial pectin powder that contained one fourth of glucose which reacted with the amino group, pectin and itself through the Maillard reaction and caramelization resulting in the elimination of water [47]. Hydrogen ions was released by the Amadori rearrangement reaction which decreased the pH of the biscuits. The addition of 0.5% commercial pectin powder in the biscuits did not observe the decrease in pH. Similar results were reported by Passos et al. [26] that the pH of biscuits decreased by 0.5 and 1.0 unit with the addition of 1% and 5% commercial pectin powder. In this study, fructose in the original formula of the biscuit was replaced by sucrose to mitigate acrylamide formation. Fructose decomposes into organic acids such as formic acid, lactic acid, levulinic acid and hydroxymethylfurfural during heating treatment [48]. The generation of organic acids decreased the pH by nearly 2 units which inhibited acrylamide formation but did not influence the sensory sensation of biscuits [9,15,49]. The addition of 0.5% chitosan also showed a decreased pH of the biscuit. However, the pH increased slightly with the addition of chitosan at the levels of 1% and 1.5% that might be due to the insolubility of chitosan and limited Maillard reaction during air frying [50].

For the moisture contents of the air fried biscuits, all the samples were below 2% and showed no significant difference. Moreover, the water activities were all below 0.3 providing considerable crispy texture to the biscuits. Katz and Labuza [51] proposed that snack foods would be crispier if a_w_ was below 0.39. In summary, the applications of hydrocolloids into biscuits did not alter the physicochemical properties of biscuits.

### 4.2. Acrylamide and HMF Levels in Biscuits

Biscuits in the control group without hydrocolloids addition contained 1340 ppb acrylamide, which were over the high levels of EU regulation (>150 ppb) [52] and Taiwan FDA suggested level (not more than 500 ppb). The applications of hydrocolloids in biscuits were found to have lower acrylamide contents. The addition of 0.5% chitosan observed the lowest acrylamide content (578 ppb in the biscuits (Figure 2A)). The biscuits with 0.5% and 1.0% pectin addition also offered lower amounts of acrylamide compared to that of the control biscuits. In the sodium-alginate-added biscuits, all samples were significantly lower than that of the control biscuit, perhaps due to the sodium cation mitigating the formation of the intermediate responsible for acrylamide formation [14]. Among the food hydrocolloids used in this study, chitosan was the most effective in terms of mitigating the formation of acrylamide. Lindsay and Jang [13] explored the possibility of using ammoniated polymers such as chitosan for blocking the carbonyl groups of neutral reducing sugars on sliced potato surfaces. They found that chitosan treatments were effected and could be more effective in combination with other acrylamide reduction technologies. Chitosan contains many amino groups which will compete with asparagine for the reaction sites of reducing sugar. Along with the lowering pH effect and slowing molecular movement effect of long chain polysaccharide, hydrocolloids tested in this study showed good mitigating effects on acrylamide formation.

Biscuits with an addition of 1.0% and 1.5% commercial citrus pectin powder were found to have HMF contents (1.31 ppb and 1.45 ppb). All the other test biscuits and control biscuits did not detect HMF. The 0.5% pectin-added biscuit did not detect HMF, which may be due to the extraction process that diluted the HMF concentrations, and it is beyond the detected limit of the UV detector. Commercial citrus pectin powder contained one fourth of glucose, and the glucose was reported to form HMF in the food system. Nguyen et al. [53] reported biscuits containing glucose or pectin detected more acrylamide formation and HMF generation. Thus, these two biscuits with high HMF contents were caused by glucose carried by commercial citrus pectin.

### 4.3. DPPH Radical Scavenging and Reducing Power of Biscuits

Many articles have demonstrated the correlation between antioxidants and acrylamide formation. However, it seems that antioxidant activity is not mainly responsible for changing the acrylamide contents [54]. In this study, the DPPH radical scavenging activities showed no significant difference among the control and hydrocolloid-added samples. It was observed that limited DPPH radical scavenging activity in 1.50% sodium-alginate-added biscuits which equaled to 110 μM ascorbic acid. The addition of hydrocolloids showed significant reducing power (Figure 3). Sodium alginate offered better reducing power over the addition of chitosan and pectin in biscuits. Chitosan showed slight reducing power (0.13–0.29) at 1 mg/mL, and moderate reducing power (0.42–0.57) at 10 mg/mL [55]. Strong reducing power (1.07) at 12.1% sulfated chitosan (11.7 × 10^4^ Da) was reported by Xing et al. [56].

### 4.4. Hardness, Color and Sensory Evaluation of Biscuits

The flavor, texture and overall acceptability of biscuits made with hydrocolloids addition were not significantly different with that of the control, except for the appearance and odor. The means of odor and appearance scores of biscuits were all over five, indicating that they were acceptable to the consumers, and the hydrocolloids additions did not contribute to the negative effect of odor. The scores of appearances of hydrocolloid-added samples were lower than the control one. However, there was no obvious difference among test biscuits (125–140) and the control (137) in the browning index (Table 2). Similar results were reported that the addition of 0.5% and 1.0% commercial pectin powder into biscuits did not alter the appearance and the color measured by Lab value [26].

A correlation analysis was evaluated among the physicochemical properties in order to better understand the relationships of different quality attributes (Appendix A). The reducing power of biscuits were positively correlated with the ascending concentrations of the added hydrocolloids, but negatively correlated with the formation of acrylamide. Therefore, the results demonstrated the potential on the mitigation effects of higher-level chitosan, pectin and sodium alginate additions (>1.5%) on acrylamide formation in biscuits.

## 5. Conclusions

This study investigated the addition of hydrocolloids such as chitosan, pectin and sodium alginate in air fried biscuits and observed the mitigating effects on acrylamide formation due to the increased reducing power and decreased pH. Pectin addition slightly decreased the appearance and odor sensory scores in biscuits whereas others showed no effect on the appearance. Biscuits with the addition of 1% sodium alginate and 0.5% chitosan significantly mitigated the formation of acrylamide, although they slightly decrease the appearance and odor acceptability. Therefore, the application of sodium alginate and chitosan addition in biscuit formulation was suggested for air frying biscuits. Food hydrocolloid demonstrated the ability of increasing reducing power in biscuits that prevented the Maillard reaction which triggered amino acid and reducing sugar to form acrylamide during air fryer baking. Among sodium alginate and chitosan, the latter one was observed to react with reducing sugar more effectively to mitigate acrylamide formation.

## Figures and Tables

**Figure 1 polymers-14-03961-f001:**
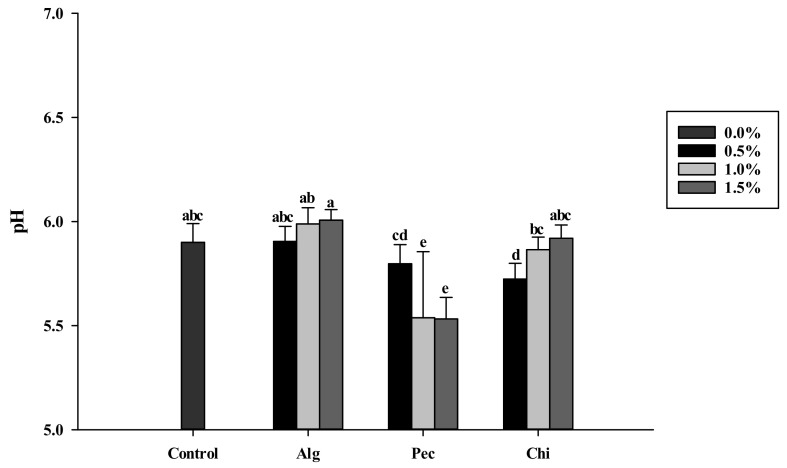
pH values of biscuits made of sodium alginate, pectin and chitosan additions in different concentrations. a, b, c, d, e: indicate statistically significant difference (p<0.05) among treatments (n=3). Alg: Sodium alginate; Pec: Pectin; Chi: Chitosan.

**Figure 2 polymers-14-03961-f002:**
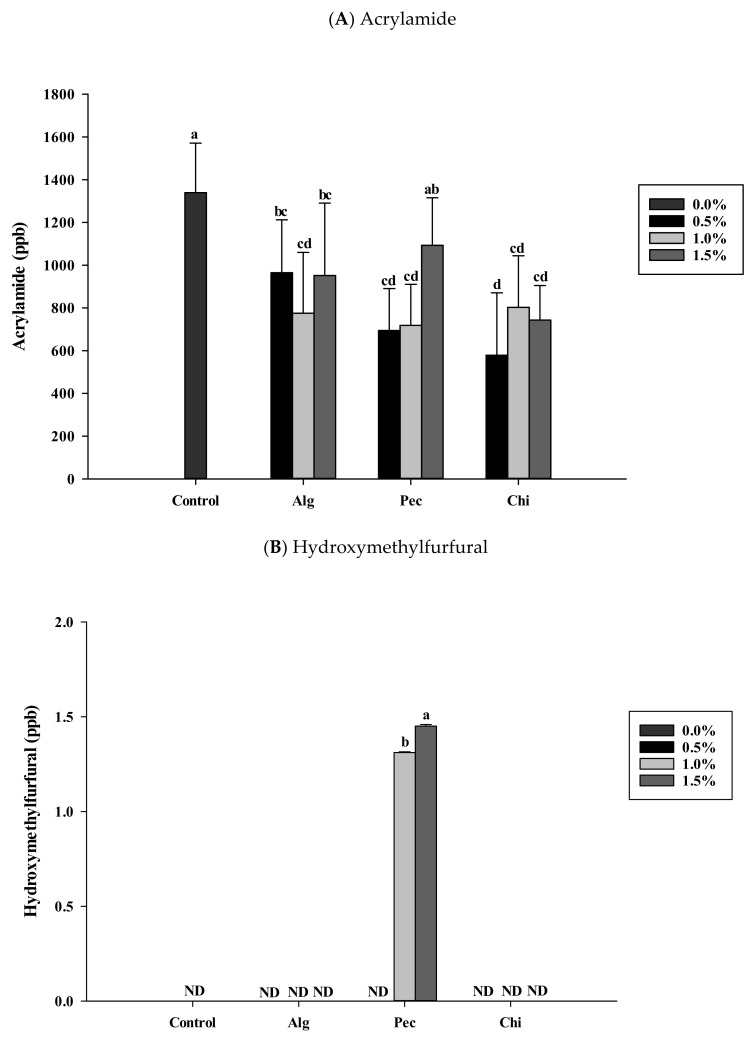
Acrylamide (**A**) and HMF (**B**) contents of biscuits added of sodium alginate, pectin and chitosan in different concentrations. a, b, c, d: indicate statistically significant difference (p<0.05) among treatments (n=3). Alg: Sodium alginate; Pec: Pectin; Chi: Chitosan; ND: Not detected.

**Figure 3 polymers-14-03961-f003:**
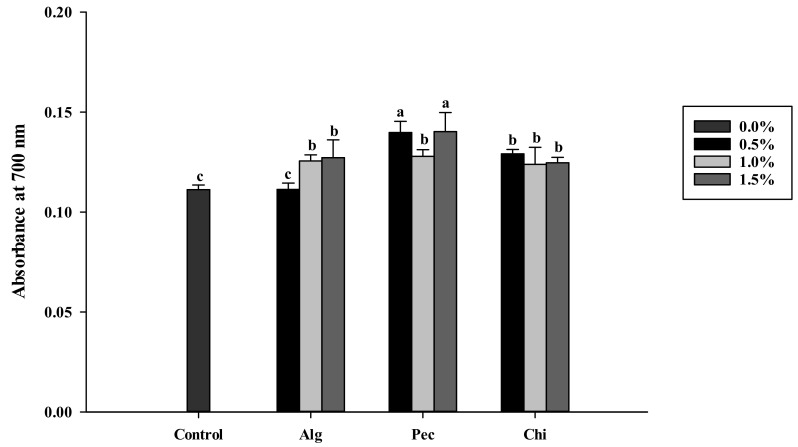
Reducing power of biscuits added of different concentrations of sodium alginate, pectin and chitosan. a, b, c: indicate statistically significant difference (p<0.05) among treatments (n=3). Alg: Sodium alginate; Pec: Pectin; Chi: Chitosan.

**Figure 4 polymers-14-03961-f004:**
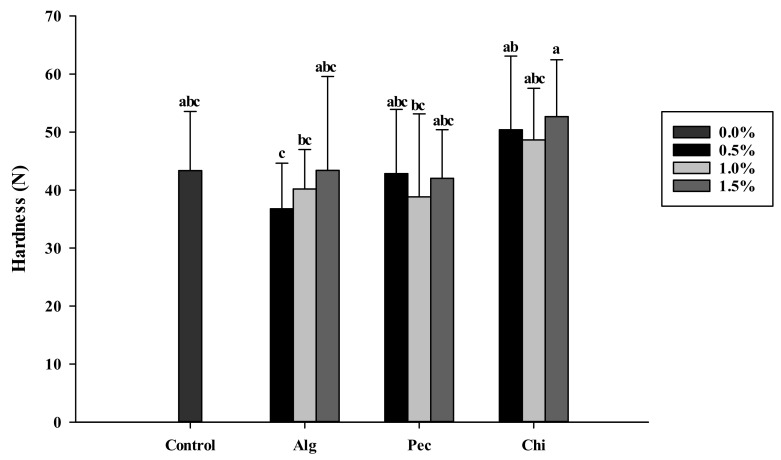
Hardness of biscuits added of sodium alginate, pectin and chitosan in different concentrations. a, b, c: indicate statistically significant difference (p<0.05) among treatments (n=3). Alg: Sodium alginate; Pec: Pectin; Chi: Chitosan.

**Figure 5 polymers-14-03961-f005:**
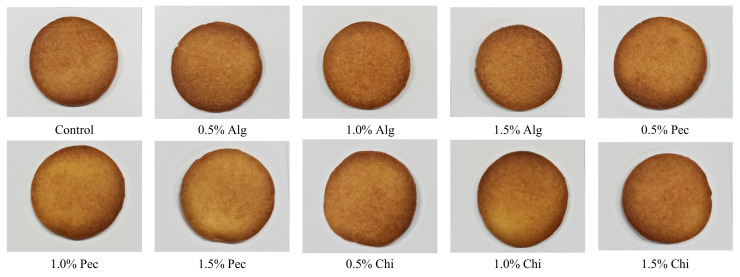
Appearance of biscuits in different formulations. Alg: Sodium alginate; Pec: Pectin; Chi: Chitosan.

**Table 1 polymers-14-03961-t001:** Diameter, thickness and spread ratio of biscuits added of sodium alginate, pectin or chitosan in different concentrations.

Groups		Diameter (mm)	Thickness (mm)	Spread Ratio
Control	0.0%	57.39 ± 1.37 ^cd^	8.54 ± 0.50 ^abc^	6.74 ± 0.39 ^bcd^
Sodium alginate	0.5%	57.06 ± 0.57 ^cd^	8.16 ± 0.62 ^c^	7.03 ± 0.58 ^abc^
	1.0%	56.54 ± 0.82 ^d^	8.66 ± 0.81 ^ab^	6.58 ± 0.59 ^d^
	1.5%	57.07 ± 0.70 ^cd^	8.24 ± 0.41 ^bc^	6.94 ± 0.36 ^abcd^
Pectin	0.5%	58.34 ± 1.17 ^ab^	8.15 ± 0.46 ^c^	7.18 ± 0.39 ^a^
	1.0%	57.95 ± 1.14 ^abc^	8.24 ± 0.45 ^bc^	7.06 ± 0.43 ^ab^
	1.5%	58.47 ± 1.06 ^ab^	8.28 ± 0.24 ^abc^	7.07 ± 0.18 ^ab^
Chitosan	0.5%	57.62 ± 1.45 ^bc^	8.67 ± 0.44 ^ab^	6.67 ± 0.43 ^cd^
	1.0%	58.80 ± 0.51 ^a^	8.59 ± 0.41 ^abc^	6.86 ± 0.33 ^abcd^
	1.5%	57.66 ± 1.26 ^bc^	8.74 ± 0.43 ^a^	6.62 ± 0.35 ^d^

Express as mean ± standard deviation (n=3). a, b, c, d: indicate statistically significant difference (p<0.05) among treatments on the same column. Spread ratio = Diameter/Thickness.

**Table 2 polymers-14-03961-t002:** Color values in sodium alginate, pectin- or chitosan-added biscuits in different concentrations.

Groups		CIE L*	CIE a*	CIE b*	ΔE	Browning Index
Control	0.0%	66.70 ± 5.27 ^a^	0.78 ± 3.26 ^a^	53.30 ± 5.11 ^ab^		137.38 ± 18.84 ^a^
Sodium alginate	0.5%	69.59 ± 5.93 ^a^	−0.68 ± 4.02 ^a^	52.65 ± 3.72 ^ab^	6.84 ± 4.97 ^ab^	125.04 ± 24.11 ^a^
	1.0%	70.00 ± 3.36 ^a^	0.02 ± 2.43 ^a^	53.15 ± 3.35 ^ab^	5.50 ± 2.70 ^ab^	124.99 ± 15.31 ^a^
	1.5%	69.80 ± 6.18 ^a^	−0.19 ± 2.44 ^a^	53.93 ± 3.36 ^ab^	5.57 ± 5.74 ^ab^	128.74 ± 12.49 ^a^
Pectin	0.5%	65.33 ± 6.02 ^a^	−0.82 ± 4.05 ^a^	50.94 ± 6.63 ^b^	8.75 ± 4.65 ^a^	132.24 ± 35.85 ^a^
	1.0%	66.43 ± 7.51 ^a^	−0.05 ± 2.65 ^a^	52.67 ± 4.46 ^ab^	7.69 ± 4.26 ^ab^	140.25 ± 43.77 ^a^
	1.5%	65.20 ± 5.88 ^a^	−0.21 ± 3.32 ^a^	49.92 ± 4.73 ^b^	8.02 ± 3.48 ^ab^	128.04 ± 24.71 ^a^
Chitosan	0.5%	68.98 ± 5.53 ^a^	0.26 ± 1.54 ^a^	56.00 ± 3.15 ^a^	5.61 ± 4.67 ^ab^	140.30 ± 7.55 ^a^
	1.0%	68.65 ± 3.34 ^a^	1.47 ± 0.94 ^a^	55.60 ± 1.87 ^a^	4.54 ± 1.76 ^ab^	140.81 ± 5.67 ^a^
	1.5%	67.64 ± 2.78 ^a^	1.60 ± 1.98 ^a^	54.29 ± 1.91 ^ab^	3.68 ± 1.71 ^b^	138.79 ± 6.55 ^a^

Express as mean ± standard deviation (n=3). a, b: indicate statistically significant difference (p<0.05) among treatments on the same column.

**Table 3 polymers-14-03961-t003:** Sensory evaluation of different hydrocolloid of biscuits.

	Appearance Acceptability	Odor Acceptability	Flavor Acceptability	Texture Acceptability	Overall Acceptability
Control	7.05 ± 1.48 ^a^	7.04 ± 1.41 ^a^	6.90 ± 1.31 ^a^	6.79 ± 1.31 ^a^	6.92 ± 1.18 ^a^
1.0% Sodium alginate	6.44 ± 1.47 ^b^	6.71 ± 1.32 ^ab^	6.78 ± 1.64 ^a^	6.83 ± 1.37 ^a^	6.96 ± 1.22 ^a^
0.5% Pectin	5.97 ± 1.75 ^b^	6.42 ± 1.34 ^b^	6.74 ± 1.48 ^a^	6.84 ± 1.69 ^a^	6.81 ± 1.27 ^a^
0.5% Chitosan	6.29 ± 1.58 ^b^	6.57 ± 1.36 ^b^	6.44 ± 1.51 ^a^	6.78 ± 1.33 ^a^	6.64 ± 1.27 ^a^

Express as mean ± standard deviation (n=77). a, b: indicate statistically significant difference (p<0.05) among treatments on the same column. 1–9 scale: 1 = dislike very much, 9 = like very much.

## Data Availability

Not applicable.

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
