# Peer review of "Effects of Sodium Alginate, Pectin and Chitosan Addition on the Physicochemical Properties, Acrylamide Formation and Hydroxymethylfurfural Generation of Air Fried Biscuits"

_polymers, 2022, doi:10.3390/polym14193961_

Round 1

Reviewer 1 Report

Reviewer comments:

This manuscript presents an interesting work about the effect of Sodium Alginate, Pectin and Chitosan Addition on 2 the Formation of Acrylamide and the Physicochemical Proper- 3
ties of Biscuits in Air Frying Biscuits.
The manuscript is very concrete, with very completes information and provides the necessary details about the field.  In general, the work is good and original; however, manuscript has some deficiencies due to language. Additionally, the reviewer has some questions, comments and suggestions to improve the manuscript:

Abstract: authors must specify that the control was made without any addition of the 3 hydrocolloids

Keywords: authors must add pectin to the list

Materials and methods

Line 102: Proximate composition of cake flour must be provided

Line 221: please correct as following:

The biscuit formulation followed the American Association of Cereal Chemists (AACC) method 10-54 (AACC, 2000) with slight modification (80.0 g cake flour, 17.6 g deionized water, 25.6 g sucrose, 0.8 g non-fat dry milk, and 32.0 g shortening)

How authors fixed the levels of each hydrocolloids: 0.5%, 1 and 1.5%.

Line 144: lease correct 1482 ×g to homogenize in whole manuscript

Results:

Authors must present the results of physicochemical properties of biscuits in a table and then interpret the table (thickness, diameter, spread ratio)

Line 225: ‘The thickness and diameter of biscuits made with food hydrocolloid addition were in the range of 8.15 mm to 8.67 mm and 56.54 mm to 58.80 mm, respectively”:

Respectively to what? and the range of these value is for  which hydrocolloids? (the order is important; Sodium Alginate, Pectin and Chitosan?), please rewrite

Authors must verify the word: cookie in some sentence (line: 230) biscuit and cookie are not exactly the same

Line 278: lease correct: not significant difference not different

Table 1: sensory evaluation

Authors used the word: texture but in materials and methods they used crisp, please correct in sensory evaluation these words are not the same

Images of different biscuits with the three hydrocolloids are important to show the colors of different biscuits comparing to control with high content of acrylamide

Conclusions: Please rewrite this sentence: unclear

“The acrylamide content of biscuits with food hydrocolloid addition was low compared to the control biscuit might due to these hydrocolloids increasing reducing powder and mitigating acrylamide formation in biscuits”.

Author Response

Responses to Comments and Suggestions for Authors

Polymers-1902076

Title: Effects of Sodium Alginate, Pectin and Chitosan Addition on the Physicochemical Properties, Acrylamide Formation and Hydroxymethylfurfural Generation of Air Fried Biscuits

Dear Reviewer #1

The authors are extremely grateful to anonymous reviewer involved for providing his/her excellent comments and valuable advice in this paper. We have revised the paper based on the reviewer’s comments. We have pleasure in requesting the reviewer to review this paper. Thank you. Your prompt attention to this paper is very much appreciated.

Comments and Suggestions for Authors

This manuscript presents an interesting work about the effect of Sodium Alginate, Pectin and Chitosan Addition on 2 the Formation of Acrylamide and the Physicochemical Properties of Biscuits in Air Frying Biscuits. The manuscript is very concrete, with very completes information and provides the necessary details about the field.  In general, the work is good and original; however, manuscript has some deficiencies due to language. Additionally, the reviewer has some questions, comments and suggestions to improve the manuscript:

Point 1: Abstract: authors must specify that the control was made without any addition of the 3 hydrocolloids.

Response 1: Thanks for your concern. We have revised the article title, and rewritten certain places in the revised manuscript. All the changes are marked as red in the revised manuscript. We added the information for the control biscuit in the abstract as”  control biscuit without hydrocolloids addition” in abstract lines 19 & 20. We asked Dr. Fang to rewrite the revised manuscript again. If it is not good enough, we will find another professional English editing company to help us. Thanks for the suggestions and we are very much appreciated your consideration on this matter. (Please see the revised manuscript).

Point 2: Keywords: authors must add pectin to the list

Response 2: The keyword "pectin" in line 25 was added to keyword section. Thanks for the suggestion.

Point 3: Materials and methods

Line 102: Proximate composition of cake flour must be provided

Response 3: The sentence “The cake flour contains moisture 13.0%, protein 8.0%, fat 1.0%, ash 0.4%, and total carbohydrate 77.6%” in lines 80-81 for proximate composition of cake flour was added. Thanks for your suggestion.

Point 4: The biscuit formulation followed the American Association of Cereal Chemists (AACC) method 10-54 (AACC, 2000) with slight modification (80.0 g cake flour, 17.6 g deionized water, 25.6 g sucrose, 0.8 g non-fat dry milk, and 32.0 g shortening)

How authors fixed the levels of each hydrocolloids: 0.5%, 1 and 1.5%.

Response 4: We added the sentence “Chitosan, pectin or sodium alginate (0% (control), 0.5% (0.4 g), 1.0% (0.8 g) or 1.5% (1.2 g)) of the cake flour weight (80 g) was used in the biscuit’s formulation. Thanks for pointing out the problems at section 2.2.

Point 5: Line 144: Please correct 1482 ×g to homogenize in whole manuscript

Response 5: We revised the format in whole manuscript in lines 123 and 151. (Please see the revised Materials and Methods at pages 3 & 4).

Point 6: Results: Authors must present the results of physicochemical properties of biscuits in a table and then interpret the table (thickness, diameter, spread ratio)

Response 6: Table 1 Diameter, thickness and spread ratio of different sodium alginate, pectin and chitosan concentration of biscuits was added in the section 3.1 of Results. Sentences in lines 207-214 were revised and the discussion sentences of the paragraph was added to section 4.1 in lines 294-300. Thanks for the suggestion.

Point 7: Line 225: ‘The thickness and diameter of biscuits made with food hydrocolloid addition were in the range of 8.15 mm to 8.67 mm and 56.54 mm to 58.80 mm, respectively”:

Respectively to what? and the range of these value is for which hydrocolloids? (the order is important; Sodium Alginate, Pectin and Chitosan?), please rewrite

Response 7: We revised sentences in lines 207-214 in section 3.1 of Results to help the reader could understand well. Thanks for reminding the problem.

Point 8: Authors must verify the word: cookie in some sentence (line: 230) biscuit and cookie are not exactly the same.

Response 8: The difference between biscuit and cookie was added in lines 294-300 of Discussion section 4.1.

Point 9: Line 278: Please correct: not significant difference not different

Response 9: The description of “not significant difference” was revised as “There was no significant difference among control and hydrocolloid added biscuits on DPPH radical scavenging activities.” Thanks for point the problems and please see the revised Results section 3.3 at page 8.

Point 10: Table 1: sensory evaluation

Authors used the word: texture but in materials and methods they used crisp, please correct in sensory evaluation these words are not the same. 

Response 10: The word “crispy” was revised to “texture” in line 193 of Materials and Methods. Please see page 5 of the revised manuscript.

Point 11: Images of different biscuits with the three hydrocolloids are important to show the colors of different biscuits comparing to control with high content of acrylamide.

Response 11: Table 2 and Figure 5 were added to show the colors of different biscuits comparing to control. Thanks for point out the problem.

Point 12: Conclusions: Please rewrite this sentence: unclear

“The acrylamide content of biscuits with food hydrocolloid addition was low compared to the control biscuit might due to these hydrocolloids increasing reducing powder and mitigating acrylamide formation in biscuits”.

Response 12: The conclusion was rewritten by Dr. Fang. Thanks for the suggestion, I am appreciated for your time and efforts on this paper.

Yours truly,

Wen-Chieh Sung, Ph.D.

Professor

Department of Food Science

National Taiwan Ocean University

Reviewer 2 Report

These are my remarks and suggestions:

 ·       The title should be modified and simplified. The authors do not investigate only acrylamide as a heat-induced contaminant but also HMF. I suggest including these in the title

·       The references should be numbered in order of appearance in square brackets. Please check Instructions for Authors https://www.mdpi.com/journal/polymers/instructions

·       The introduction part should be reviewed. However, the introduction should be simplified (it is difficult to follow). There should better explain the originality of this paper, also adding the influence of the biopolymers used on the physicochemical properties of biscuits

·       Lines 50-54. The information is not relevant here

·       Lines 230-231. Please rephrase

·       Line 234. aw instead of Aw

·       Lines 243-245. Please rephrase

·       Lines 243-245. Please rephrase. I do not understand the purpose of this statement here

·        In Figure 2, the acrylamide and HMF content are expressed as ppb and in the text as ng/g; please uniformize

·       Lines 230-231. The information should be inserted in the discussion part

·       I suggest that authors should provide the color and texture parameters obtained for the biscuit samples

·       Lines 230-231. The information should be inserted in the discussion part

·       Lines 315-316. Use a newer reference

·       Lines 376. Please provide the correlation analysis made

·       This work focused on the demonstration of experimental results; the discussions should be supplemented and solid and compared with the other works

·       Overall, it is not easy to follow the logical flow of the manuscript

Author Response

Responses to Comments and Suggestions for Authors

Polymers-1902076

Title: Effects of Sodium Alginate, Pectin and Chitosan Addition on the Physicochemical Properties, Acrylamide Formation and Hydroxymethylfurfural Generation of Air Fried Biscuits

Dear Reviewer #2

The authors are extremely grateful to anonymous reviewer involved for providing his/her excellent comments and valuable advice in this paper. We have revised the paper based on the reviewer’s comments. We have pleasure in requesting the reviewer to review this paper. Thank you. Your prompt attention to this paper is very much appreciated.

Comments and Suggestions for Authors

Point 1: These are my remarks and suggestions: The title should be modified and simplified. The authors do not investigate only acrylamide as a heat-induced contaminant but also HMF. I suggest including these in the title.

Response 1: We have revised and added HMF to the title as red marked texts in the revised manuscript. Thanks for the suggestions and we very much appreciate your consideration on this matter. (Please see the revised manuscript).

Point 2: The references should be numbered in order of appearance in square brackets. Please check Instructions for Authors https://www.mdpi.com/journal/polymers/instructions

Response 2: The cited references in the texts were numbered in order of appearance in square bracket format in revised manuscript. Thanks for pointing out the problem and suggestions.

Point 3: The introduction part should be reviewed. However, the introduction should be simplified (it is difficult to follow). There should better explain the originality of this paper, also adding the influence of the biopolymers used on the physicochemical properties of biscuits

Response 3: We asked Dr. Fang to rewrite the revised manuscript including introduction section, explain the originality of the manuscript and add the influence of the biopolymers used on the physicochemical properties of biscuits. Thanks for your suggestion.

Point 4: Lines 50-54. The information is not relevant here.

Response 4: We removed the not relevant information and rewritten the paragraph. Thanks for pointing out the problems at the Introduction section.

Point 5:  Lines 230-231. Please rephrase

Response 5: We revised sentences in lines 207-214 in section 3.1 of Results to help the reader could understand well. Thanks for reminding the problem. (Please see lines 207-214 of the revised Results at page 5).

Point 6:  Line 234. aw instead of Aw

Response 6: AW was revised to aw in lines 220 and 320. Thanks for pointing out the mistake.

Point 7:  Lines 243-245. Please rephrase

Response 7: We revised sentences in lines 216-217 in section 3.1 of revised manuscript to help the reader could understand well. Thanks for reminding the problem.

Point 8:  In Figure 2, the acrylamide and HMF content are expressed as ppb and in the text as ng/g; please uniformize.

Response 8: The unit for acrylamide and HMF levels in the text was revised to ppb in the whole manuscript. Thanks for point out the problem.

Point 9: Lines 230-231. The information should be inserted in the discussion part

Response 9: The information of biscuit thickness, diameter and spread ratio was moved to the section 4.1 of discussion. Thanks for point the problems and please see the revised Discussion 4.1 at page 11.

Point 10: I suggest that authors should provide the color and texture parameters obtained for the biscuit samples. 

Response 10: Table 2 and Figure 2 for the color and texture parameters for biscuits were added to the revised manuscript. Please see pages 9 and 10 of the revised manuscript.

Point 11: Lines 315-316. Use a newer reference.

Response 11: The cited reference, Belitz et al., 2009 was renewed to Kontogiorgos, 2021. Please see the reference #47 of revised manuscript. Thanks for the suggestion.

Point 12: Please provide the correlation analysis made

  • This work focused on the demonstration of experimental results; the discussions should be supplemented and solid and compared with the other works. Overall, it is not easy to follow the logical flow of the manuscript

Response 12: The correlation analysis of sodium alginate, pectin, and chitosan concentration, acrylamide content, pH value, DPPH radical scavenging activity, reducing power and CIE color value of biscuits were provided at supplementary Tables S1 to S3 of the revised manuscript. Thanks for the suggestion.

Yours truly,

Wen-Chieh Sung, Ph.D.

Professor

Department of Food Science

National Taiwan Ocean University

Round 2

Reviewer 1 Report

Dear Authors, I reviewed the revised version of the manuscript. This version of the manuscript followed all the recommendations suggested by reviewers to improve its redaction and quality. For this reason, I considered that this manuscript can be accepted for its publication in this journal.

Reviewer 2 Report

The manuscript has been improved.